# Bioactive Compounds: Natural Defense Against Cancer?

**DOI:** 10.3390/biom9120758

**Published:** 2019-11-21

**Authors:** Shonia Subramaniam, Kanga Rani Selvaduray, Ammu Kutty Radhakrishnan

**Affiliations:** 1Pathology Division, School of Medicine, International Medical University, Bukit Jalil, Kuala Lumpur 50050, Malaysia; sonivalai@gmail.com; 2Product Development and Advisory Services, Malaysian Palm Oil Board, Kajang, Selangor 43000, Malaysia; krani@mpob.gov.my; 3Jeffrey Cheah School of Medicine and Health Sciences, Monash University Malaysia, Bandar Sunway, Selangor 47500, Malaysia

**Keywords:** bioactive compounds, anticancer, curcumin, myricetin, geraniin, tocotrienols

## Abstract

Cancer is a devastating disease that has claimed many lives. Natural bioactive agents from plants are gaining wide attention for their anticancer activities. Several studies have found that natural plant-based bioactive compounds can enhance the efficacy of chemotherapy, and in some cases ameliorate some of the side-effects of drugs used as chemotherapeutic agents. In this paper, we have reviewed the literature on the anticancer effects of four plant-based bioactive compounds namely, curcumin, myricetin, geraniin and tocotrienols (T3) to provide an overview on some of the key findings that are related to this effect. The molecular mechanisms through which the active compounds may exert their anticancer properties in cell and animal-based studies also discussed.

## 1. Introduction

Cancer is one of the leading causes of death in the world. Cancer burden is measured based on cancer incidence and mortality. The International Agency for Research on Cancer (IARC) reported the 5-year global cancer prevalence of worldwide burden of 27 cancers for the year 2008 to be 28.8 million with 12.7 million new cancer cases and 7.6 million cancer deaths [1]. Some of the major cancer cases reported were lung (1.61 million), breast (1.38 million) and colorectal (1.23 million) cancers [2]. In addition, it was estimated that there would be 14.1 million new cancer cases and 8.2 million deaths in 2012 worldwide [3]. According to a recent global cancer statistic, there will be 18.1 million new cancer cases and 9.6 million cancer deaths [4], with lung cancer being the leading cause of death followed by breast, colorectal, stomach and liver cancer (Table 1) [4].

Cancers could result from inflammatory processes that are driven by rapid growth of intrinsic (self) origin. Some of the common hallmark features of cancers are shown in Figure 1. These include the ability of cancer cells to (i) evade apoptosis [5]; (ii) induce angiogenesis [6]; (iii) replicate limitlessly [7,8]; (iv) produce growth signals that are self-sufficient [9]; (v) be insensitive to anti-growth signals [10]; and (vi) invade tissue and metastasis [11,12]. These attributes allow the cancer cells to have limitless growth, prolonged survival and the ability to invade tissues. If these processes are not inhibited, the cancer cells can continue to grow and invade and eventually kill the cancer patient.

At present, various therapeutic approaches such as surgery, chemotherapy drugs and/or radiation are used to treat cancers. Whilst the chemotherapeutic drugs used in the treatment of cancer can provide temporary relief to the cancer patients and help prolong their life [13,14,15], many of these drugs exhibit side-effects [16,17].

## 2. Anti-Tumor Immune Responses

Activation of host immune system is a natural way for cancer patients to fight this disease. Several studies have shown that cells of the immune system can recognize and destroy tumor cells [18]. The process through which the immune system carries out this function is known as immunosurveillance [19]. As shown in Figure 2, continuous immunosurveillance takes place in the body to help the immune system to deal with “rogue” or abnormal cells. The outcome of this response is regulated by a process known as immunoediting. Cancer immunoediting refers to the dual role played by the immune system in host protection and promotion of tumor growth. Cancer immunoediting consists of three phases, which are elimination, equilibrium and escape [20,21]. If the immune system is appropriately activated, tumor growth can be inhibited and they can be destroyed. However, in some situations, the immune system may can promote tumor progression through chronic inflammation [22] and/or suppression of anti-tumor immune responses [23].

During the elimination phase (Figure 2), the innate and adaptive arms of the host immune system work hand-in-hand to destroy cancer cells before these cells can be clinically detected [24]. Many effector T-lymphocyte subsets and cytokines play key roles in eliminating tumor cells. Tumor cells that cannot be destroyed in the elimination phase can enter the equilibrium phase. The main role of the equilibrium phase is to prevent outgrowth of the tumor by enabling editing of tumor immunogenicity. In addition, T-helper-1 (Th1) cells as well as some of the cytokines that these cells produce (e.g., interleukin-12 (IL-12) and interferon-gamma (IFN-γ)) help to maintain tumor cells in a state of immune-mediated dormancy. However, maintaining immune cells constantly in this phase may allow emergence of unstable tumor cells that can overcome some of the barriers imposed by the anticancer immune responses. One of the reasons for this could be expression of new molecules on the tumor cells due to mutations, which are no longer recognized by the receptors of these lymphocytes [25]. In addition, the tumor cells may secrete mediators that could induce an immunosuppressive state within the tumor microenvironment [26]. When this happens, the tumor cells are no longer susceptible to the host immune system, enabling them to avoid the elimination and equilibrium phases and enter the escape phase. In the escape phase, tumor progression is no longer blocked by the host immune system and the tumor can be detected clinically [27].

## 3. Plant-Derived Active Compounds and Their Mechanism of Action

There are several studies which have reported on various natural bioactive compounds that have anticancer [28,29,30] and/or immune-modulating effects [31,32]. Some of these anticancer agents possess mutagenic, teratogenic and/or oncogenic properties, which can impair antibody synthesis and also cell-mediated immune responses [33]. In the scientific literature, there is an increasing number of reports which show that many phenolic compounds have potential inhibitory effects on cancer invasion and metastasis [34,35,36,37]. A number of plant-based bioactive compounds with anticancer activities have been identified in the past decade (Table 2).

In this short review, the anticancer effects of four bioactive compounds (curcumin, myricetin, geraniin and tocotrienols) will be discussed. For this review, published papers reporting on the anticancer effects of these four bioactive compounds (curcumin, myricetin, geraniin and tocotrienols) that are indexed in PubMed and/or Google Scholar were selected to be included in this review. These four bioactive compounds were chosen for this review as these compounds are the putative anticancer natural products that we are currently working with to develop bioactive cocktails that have more potent anticancer activities.

## 4. Curcumin

Diferuloylmethane, better known as curcumin (Figure 3) is major bioactive compound derived from an East-Indian plant known as *Curcuma longa*. This plant is native to the Southeast Asian region and belong to the Zingiberacae family [60]. Curcumin consists of curcuminoids compounds, which is made up from several chemicals such as curcumin, desmethoxycurcumin and bis-demethoxycurcumin [61]. Approximately 2–5% curcumin in turmeric is responsible for the yellow color as a flavoring and coloring agent in foods.

Curcumin was found to have low bioavailability due to insufficient absorption and fast elimination from the body, which was one of the limitations of this bioactive compound. Researchers have used several approaches to increase the bioavailability of curcumin including nanoparticles [62], piperine [63], phospholipid complexes [64] and liposomes [65]. Synthetic analogs of curcumin and polyphenolic curcumin analogs have been shown to have inhibitory effects against mushroom tyrosinase [66]. Among the 61 reported curcumin compounds, four compounds (E10, F10, FN1 and FN2) were reported to inhibit prostate, pancreas and colon cancer cells with IC_50_ lower than 1 µM [67]. Curcumin has been shown potent anticancer properties on human cancers including lung, pancreatic, melanoma, prostate, head and neck, breast, colorectal and ovarian cancer [68,69,70,71,72,73,74,75]. Curcumin exerts anticancer effects through several mechanisms, which affect regulation of cell growth and apoptosis. For instance, curcumin can inhibit angiogenesis [76] as well as inhibit their proliferation and metastasis [77], decrease chronic inflammation [78] and combat mutated cancer cells [77]. Bisdemethoxycurcumin showed excellent inhibitory effects with an IC_50_ value of 23.0 µM whilst the D_2_ analog showed potent inhibitory effects at 8.2 µM [79]. A curcumin analog, namely CUR3d, inhibited proliferation of liver cancer cells at 100 µmol/L, which was reported to be due to downregulation of PI3K/Akt and inhibition of the NFκB pathway, which is responsible for cancer cell growth [80]. Another curcumin analog, WZ35, was reported to have potent cytotoxic effects on prostate cancer cells with a very low IC_50_ value (2.2 µM) when compared with curcumin (20.9 µM) [81]. In another study, curcumin at 10 μM induced apoptosis in MCF-7 human breast cancer cells, which was reported to take place via the expression of wild type p53 [82]. Exposure to curcumin, increased expression of p53 and Bax, which triggered apoptosis in these cells. In a xenograft mouse model, it was shown that a low dose of curcumin (20 μg/kg) reduced the progression of breast cancer [83]. In another study, supplementation of curcumin (1 g/kg) significantly inhibited growth and metastasis to liver of colorectal cancer cells [84]. Similar anticancer effects were also reported in aggressive papillary thyroid carcinoma, where a dose-dependent effect of curcumin was reported. Higher concentrations of curcumin (12.5, 25, 50 and 100 µM) inhibited migration of K1 papillary thyroid cancer cells by downregulating metalloproteinase-9 (MMP-9) expression [85].

## 5. Myricetin

Myricetin (3,5,7,3′,4′,5′-hexahydroxyflavone cannabiscetin) is a bioflavonoid (Figure 4) widely found in food sources such as vegetables, tea, berries, red wine and medicinal plants. It was first isolated from the bark of the *Myrica nagi* Thunb, Myricaceae in 1896 with molecular formulae of C_15_H_10_O_8_ [86]. Myricetin has been credited for its therapeutic effects in cardiovascular disease [87], cancer [88], and diabetes mellitus [89,90]. Being lipophilic compounds, myricetin has poor solubility in water but can be solubilized in organic solvents such as acetone, dimethylformamide, dimethylacetamide and tetrahydrofuran.

Myricetin is stable at pH 2 and its degradation depends on pH and temperature [91]. A recent study showed that microemulsion formulation can improve the solubility of myricetin 1225 times greater than water and also enhance its anti-proliferative activity against human liver cancer cells (HepG2) [92]. Myricetin is a promising anti-carcinogen and chemo preventive agent with therapeutic potential reported in ovarian [93], colon [94], skin [95], liver [96] and breast [97] cancers. Cell-based studies have shown that myricetin inhibited proliferation of T24 bladder cancer cells by inducing cell cycle arrest at the G2/M phase by downregulating cyclin B1 and cyclin-dependent kinase cdc2 [98]. In addition, myricetin induced apoptosis in T24 cells by modulating Bcl-2 family protein and activating caspase 3 pathways. Similar findings (i.e., cell cycle arrest and induction of apoptosis) were observed in cervical cancer cells following combination treatment with myricetin (60 μM), methyl eugenol and cisplatin [99]. In a rat model, tumor progression was inhibited when the rats were fed with 100 mg/kg of myricetin, which was found to be due to inhibition of the p21 activated kinase-1 (PAK1) [100]. A recent study showed that myricetin may exert anti-metastatic effects by downregulating the expression of MMP2 and/or MMP9 in breast cancer cells [101].

## 6. Geraniin

Geraniin is a dehydroellagitannin (Figure 5) found in geraniums and regarded as main active compounds in various medicinal plants. It was first identified from *Geranium thunbergii* [102] and belongs to the Sapindaceae, Gereniaceae, Nymphaeaceae and Elaeocarpaceae families [102,103]. Geraniin has been credited to possess high antioxidant, antibacterial, anti-hyperglycemic, anti-viral and anticancer activities [104,105,106,107,108]. The hydrolyzed compounds from geraniin were identified as gallic acid, corilagin, and ellagic acid. As shown in Figure 5, geraniin contains galloyl groups with additional hydroxyl structure to ortho-dihydroxy groups, which have high nitrogen oxide (NO) scavenging ability. Corilagin and gallic acid contain galloyl group also contribute to the intrinsic antioxidant activities of geraniin [109].

Geraniin isolated from fruit of Emblica (*Phyllantus emblica* L.) was found to have an anticancer effect on MCF-7 human breast cancer cells [110]. Cell-based assays using murine splenocytes showed that geraniin inhibited proliferation of MCF-7 human breast cancer cells with IC_50_ value of 13.2 μg/mL [110]. Similarly, geraniin extracted from *Phyllanthus urinaria* Linn was reported to have anti-proliferative and pro-apoptotic effects on MCF-7 cells with IC_50_ value 9.94 µM [111]. Geraniin triggered apoptosis by activating the p38 MAPK signaling pathway [111]. Epithelial–mesenchymal transition (EMT) is reported to play an important role in cancer metastasis [112]. Geraniin inhibited transforming growth factor beta-1 (TGF-β-1)-induced EMT in lung cancer cells by increasing the expression of E-cadherin and inhibiting expression of Snail, a transcription factor crucial for induction of EMT [113]. In addition, activation of Smad-2 was inhibited in TGF-β-1-induced EMT, suggesting that geraniin may play a role in preventing metastasis and EMT in TGF-β-1-induced signaling pathway [113].

## 7. Tocotrienol

Vitamin E encompass two major class of fat-soluble antioxidants namely tocopherols and tocotrienols (T3) [114]. There are eight dietary components identified to be a member of the vitamin E family, which are tocopherols (α, β, γ, δ) and tocotrienols (α, β, γ, δ) [115]. The chemical structure of tocotrienols and its various isoforms are shown in Figure 6.

The major sources of dietary tocopherols are plant oils such as wheat-germ oil, safflower-seed oil, maize oil, soya bean oil [116], whilst the main sources of tocotrienol are palm oil, rice bran oil, and palm kernel oil [117]. Tocotrienols are main phytonutrients found in palm oil and can be found in the palm oil fraction known as tocotrienol-rich fraction (TRF) [118]. TRF contains three main isoforms of tocotrienol, which are αT3 (29%), γT3 (28%) and δT3 (14%) isomers [119]. Tocotrienols are reported to possess anti-thrombotic [120], antioxidant [121], neuroprotective [122] and cardio-protective [123] activities as well as immune modulatory [124,125] properties. Both cell-based and experimental model studies have suggested that tocotrienols also possess anti-tumor properties as these compounds can inhibit proliferation of many cancer cell lines including prostate [126], breast [127], skin [128], colon [129], stomach [130], pancreatic [131], liver [132] and lung [133] cancers. The anticancer effects induced by tocotrienol are reported to be mediated through apoptosis [134], anti-angiogenesis [135], anti-proliferative [136] and/or immunoregulation [125]. Tocotrienol isoforms inhibited proliferation of human breast cancer cells in the following order: αT3 < TRF < γT3 < δT3 [137]. In addition, daily supplementation of 1 mg of TRF was reported to inhibit tumor growth in a syngeneic murine model of breast cancer [31,124,137]. A similar observation was also reported in a xenograft athymic mouse model of breast cancer where a significant delay in the onset of breast cancer in mice fed with 1 mg TRF was observed [138]. This delay was reported to be due to down-regulation of the c-myc oncogene in the breast cancer cell and upregulation of the CD59 glycoprotein precursor gene, which was responsible for immune regulation. In another study, supplementation with δT3 inhibited proliferation and migration of lung cancer cells in a dose- and time-dependent manner [139]. This inhibition was due to inhibition of NFκB activity and signaling via the NOTCH-1 pathway by δT3. In another study, daily supplementation of 1 mg TRF was reported to inhibit growth of breast cancer in mice as well as reduce the levels of vascular endothelial growth factor (VEGF) in serum [135]. In addition, murine breast cancer cells (4T1) cells treated with TRF or δT3 were found to induce marked inhibition of IL-8 and VEGF genes, which play important roles in tumor development [140]. These findings suggest that TRF also possesses anti-angiogenesis activity. In the same mouse model of breast cancer, it was shown that daily supplementation with TRF may exert anticancer effects by upregulating the expression of the IL24 gene [140].

## 8. Conclusions

Natural products have the potential to serve as chemotherapeutic as well as chemopreventive agents in the treatment of cancer. The bioactive compounds derived from many natural plant sources could be a possible means to provide protection against cancer or used as a treatment approach against cancer. Curcumin and tocotrienols show much promise to be developed as chemopreventive and/or novel therapeutic agents in the fight against cancer as there are many studies that show that these bioactive agents possess potent anticancer activities. Although there are some studies that have demonstrated how these compounds exert anticancer effects, the exact target remains elusive. Hence, more work needs to be carried out to know to understand exactly how these compounds act as this information would be useful in developing therapeutic cocktails made up of various bioactive agents that can target different molecules to produce better therapeutic effects.

## Figures and Tables

**Figure 1 biomolecules-09-00758-f001:**
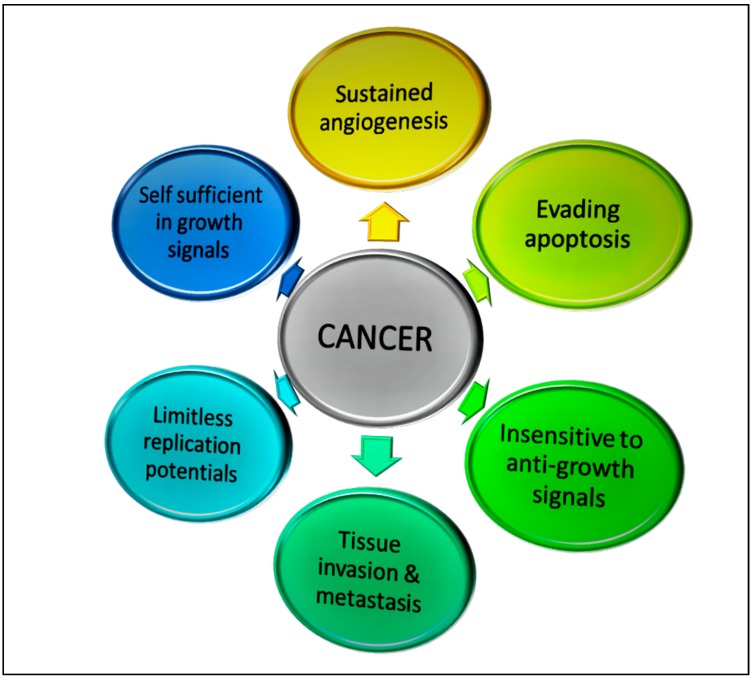
Hallmark features of tumors that allow them to grow uncontrollably and metastasize.

**Figure 2 biomolecules-09-00758-f002:**
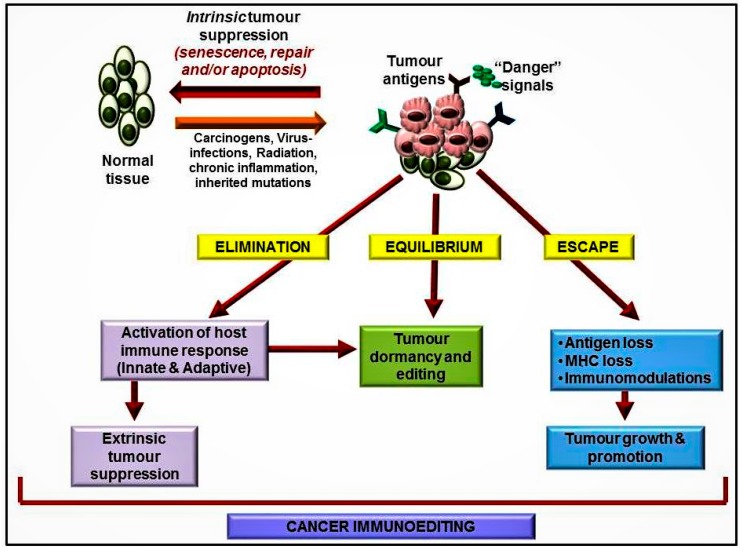
The three phases of cancer immunoediting: elimination, equilibrium and escape.

**Figure 3 biomolecules-09-00758-f003:**
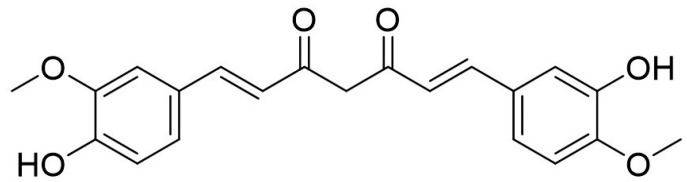
Chemical structure of Curcumin.

**Figure 4 biomolecules-09-00758-f004:**
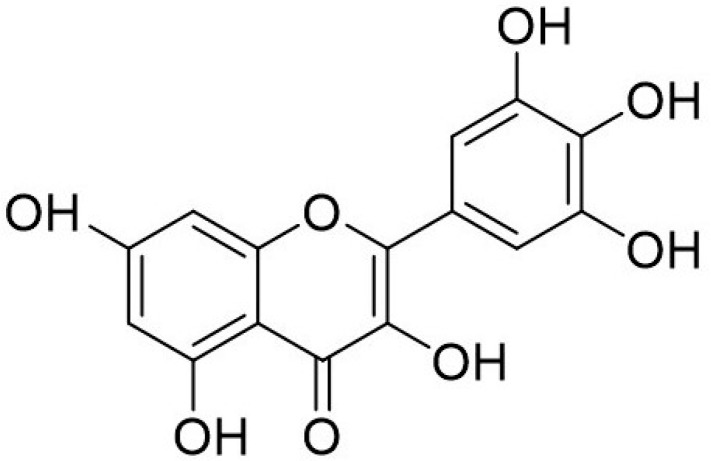
Chemical structure of Myricetin.

**Figure 5 biomolecules-09-00758-f005:**
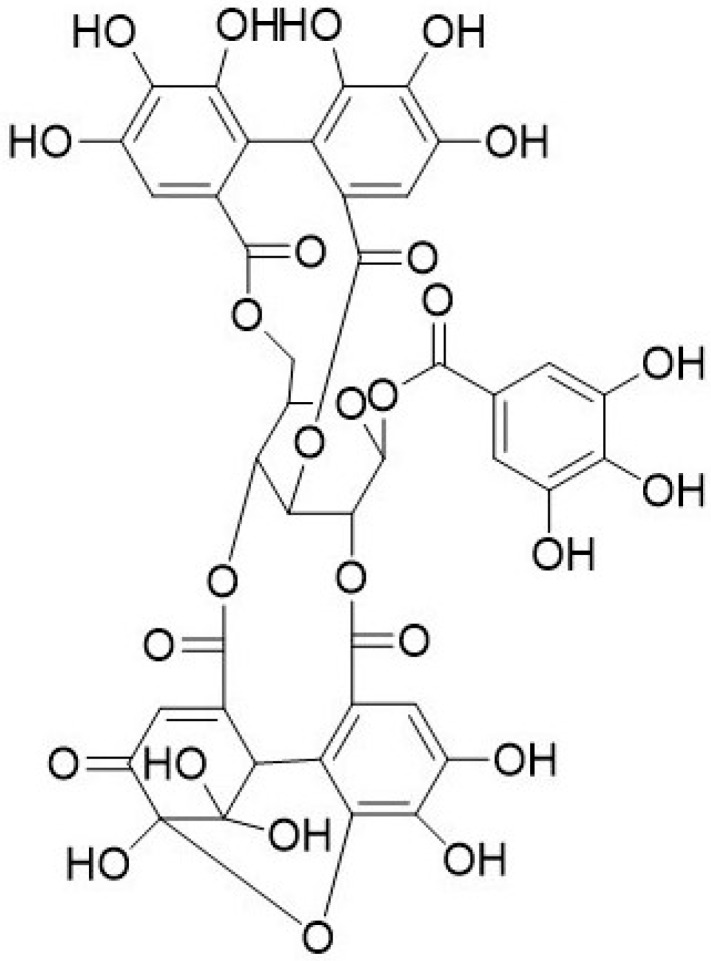
Chemical structure of geraniin.

**Figure 6 biomolecules-09-00758-f006:**
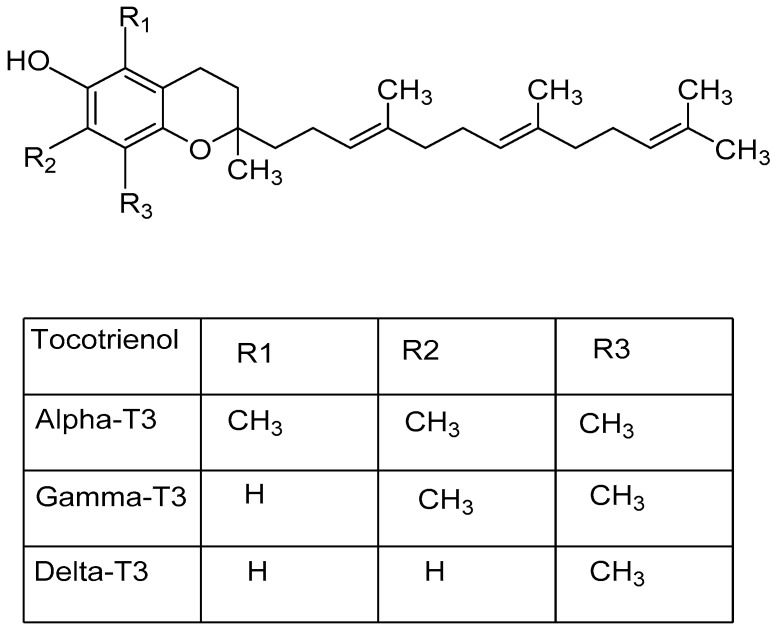
Chemical structure of tocotrienols and its four major isoforms. (T3: Tocotrienol).

**Table 1 biomolecules-09-00758-t001:** Cancer global statistics 2018.

Cancer Types	Deaths	New Cases
Lung	1.76 million	2.09 million
Stomach	782,685	1.03 million
Liver	781,631	841,080
Breast	626,679	2.02 million
Colon	551,269	1.09 million
Esophagus	508,585	572,034
Pancreas	432,242	458,918
Prostate	358,989	1.27 million
Source: [4]		

**Table 2 biomolecules-09-00758-t002:** Anticancer activities of selected natural bioactive compounds.

Target Cancer	Compounds	Biological Activity	Dosage/Concentration	Ref.
Breast	Fucoxanthin	Anticancer	10 µM	[38]
	Punicalagin	Anticancer	10 mg/mL	[39]
	Curcumin	Apoptosis	5–50 µg/mL	[40]
Lung	Anthocyanin	Anti-proliferative	400 μg/mL	[41]
	Triterpenoids	Anticancer	22.4 μmol/L	[42]
	Saponin	Anticancer, apoptosis	50 μg/mL	[43]
Pancreatic	Genistein	Anticancer	60 µM	[44]
	Garcinol	Anti-proliferative	7 μM	[45]
	Limonoids	Anti-proliferative	18–42 µM	[46]
	Crocin	Apoptosis	10 g/L	[47]
Colorectal	Carotenoids	Anti-proliferative	250 μg/mL	[48]
	Β-sitosterol	Anticancer, apoptosis	266.2 μM	[49]
	Saponin	Anticancer	5, 10 or 20 mg/kg	[50]
	Genistein	Anti-proliferative	50 μM	[51]
Prostate	Gallic acid	Anticancer	100 µg/mL	[52]
	Neobavaisoflavone, psoralidin	Apoptosis	50 µM	[53]
	Rhodioflavonoside	Apoptosis	80 µg/mL	[54]
Ovarian	Corilagin	Apoptosis	20–40 μM	[55]
	Gallic acid	Anti-proliferation	40 µM	[56]
	Ellagic acid	Anti-metastasis	50 mg/kg	[57]
Blood	Epigallocatechin gallate	Apoptosis	3–25 μg/mL	[58]
	Rosavin	Anticancer	8 µg	[59]

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
