# Peer review of "Bioactive Compounds: Natural Defense Against Cancer?"

_biomolecules, 2019, doi:10.3390/biom9120758_

Round 1
Reviewer 1 Report
The manuscript entitled “Bioactive Compounds: Natural Defense Against Cancer?” by Subramaniam et al. report available literature in the area of natural product derived bioactive compounds for use in natural defense against cancer. Authors selected only few plant-based bioactive compounds such as curcumin, myricetin, geraniin and tocotrienols to cover in this review article. The author cited about 128 citations and drew few figures and tables. The manuscript seems a short review article that lacks numerous information related to the presented selected molecules such as the active dosages of bioactive compounds, their formulations, and existed or developed synthetic analogs that need to include in the manuscript. There are several typos, and correction needs to follow before the manuscript could be further considered to publish in this high impact journal (“Biomolecules, IF ~4.6).
Minor comments:
Figure 2 and Table 2 caption need to re-write more appropriate. Figure 3 structure need to re-draw as methoxy, carbonyl, and phenolic OH not correctly draw. Similarly, other figures need to redraw (Figure 4). Figure 5 need to redraw to show missing stereochemistry. Please use word anti-cancer as anticancer throughout the manuscript. Draw a figure /table to list various applications of curcumin and its limitation or use with drug delivery application. The authors should provide more details about the curcumin dosages, activity, and details in the review.Author Response
Please see the attachment

Reviewer 2 Report
The manuscript fits within the scope of the journal. The manuscript is interesting and the idea is very nice. The revisions are necessary to improve clarity of the presentation and needed to make convincing scientific arguments. Therefore, I recommend publication of the manuscript only after major revisions.
I have some recommendations for authors:
- What was the working method? Even if it is a review, the working method must be specified in abstract/introduction.
- Please highlight the aim and degree of novelty and originality of the work.
- It is not clear why did the authors choose this theme.
- Please extend the references with recent literature data. In this sense, the use of relevant MS published in the last years and focused on bioactivity of plant extract (for ex: doi.org/10.1080/10715760310001625609, doi.org/10.1007/s00226-018-1071-5, doi.org/10.3390/medicines5030089) could also be of great interest. Currently, further valuable information to the readers is needed, in order to offer a whole vision of the issue.
- Line 78, 89, 17 - Please rewrite the figures and table title. They seem incomplete. Please analyze!
- Table 2 – In column ”Compounds” you have included the scientific names of the plants, which is not correct according to the title of the column.
- Please use the scientific name of the plant, uniformly. Full scientific name, including paternity (e.g. Margaritaria discoidea (Baill.) G.L.Webster).
- Rearrange the whole text, because it is difficult to understand.
- Include in the text potential research directions.
- A clearer presentation of the conclusions of the paper is necessary.
- Extensive editing of English language and style required
Round 2
Reviewer 1 Report
Author revised the manuscript appropriately in the light of reviewers comments. However, some minor correction noted as below:
Table 1 Caption need to start with capital first word e.g. Cancer
Table 2 should contain another column which show the dosage/concentration of bioactive compound produces activity. This will help reader to comprehend more information related to bioactivity of compound.
Reviewer 2 Report
The revised manuscript has addressed the issues raised in the previous review. The manuscript is now suitable for publication.
Author Response
Reviewer 2 Comment:
The revised manuscript has addressed the issues raised in the previous review. The manuscript is now suitable for publication.
Author's response:
Thank you.